

# Ursolic acid attenuates oligospermia in busulfan-induced mice by promoting motor proteins

Jin Dong[1,2,3,*], Taowen Ye[4,*], Yanli Dong[4], Jie Hui[5] and Xiaorong Wang[1,2,3]

[1] Center for Reproductive Medicine, Affiliated Maternity and Child Health Care Hospital of Nantong University, Nantong, Jiangsu, China
[2] Nantong Institute of Genetics and Reproductive Medicine, Affiliated Maternity and Child Health Care Hospital of Nantong University, Nantong, Jiangsu, China
[3] Nantong Key Laboratory of Genetics and Reproductive Medicine, Nantong, JIangsu, China
[4] Institute of Reproductive Medicine, Medical School, Nantong University, Nantong, Jiangsu, China
[5] Lianyungang Higher Vocational Technical College of Traditional Chinese Medicine, Lianyungang, Jiangsu, China
[*] These authors contributed equally to this work.

Corresponding author
Xiaorong Wang, xr0104@ntu.edu.cn

## ABSTRACT

**Background**. Oligospermia is one of the most common reasons for male infertility which is troubling numerous couples of child-bearing age. This investigation scrutinizes the implications and mechanistic underpinnings of ursolic acid's effect on busulfan-induced oligospermia in mouse models.

**Methods**. A singular intraperitoneal injection of busulfan at a dosage of 30 mg/kg induced oligospermia. Two weeks subsequent to this induction, mice were subjected to various dosages of ursolic acid (10, 30, and 50 mg/kg body weight, respectively) on a daily basis for four consecutive weeks. Following this treatment period, a meticulous analysis of epididymal sperm parameters, encompassing concentration and motility, was conducted using a computer-assisted sperm analysis system. The histopathology of the mice testes was performed utilizing hematoxylin and eosin staining, and the cytoskeleton regeneration of the testicular tissues was analyzed *via* immunofluorescent staining. Serum hormone levels, including testosterone, luteinizing hormone, and follicle-stimulating hormone, as well as reactive oxygen species levels (inclusive of reactive oxygen species and malondialdehyde), were gauged employing specific enzyme-linked immunosorbent assay kits. Differentially expressed genes of testicular mRNA between the oligospermia-induced group and the various ursolic acid treatment groups were identified through RNA sequencing analysis.

**Results**. The results revealed that a dosage of 50 mg/kg ursolic acid treatment could increase the concentration of epididymal sperm in oligospermia mice, promote the recovery of testicular morphology, regulate hormone levels and ameliorate oxidative damage. The mechanism research results indicated that ursolic acid increased the expression level of genes related to motor proteins in oligospermia mice.

## INTRODUCTION

Currently, infertility affects nearly 15% of couples of childbearing age around the world and approximately half attributable to male factors (*Agarwal, Hamada & Esteves, 2012*; *Levine et al., 2017*; *Zhao et al., 2022*). About 40% of male infertility causes cannot be identified, called idiopathic male infertility (*Bracke et al., 2018*; *Sengupta et al., 2022*), including teratospermia, azoospermia, asthenospermia, and oligospermia (*Lu & Huang, 2012*). Oligospermia is a prevalent symptom of male infertility. Surgical intervention and sophisticated auxiliary technology serve as treatment modalities for some oligospermia patients, albeit their exorbitant cost coupled with suboptimal outcomes (*Jiang et al., 2017*). In clinical practice, some patients with oligospermia choose medication treatment through empirical therapy, while the therapeutic effect is still ideal (*Yang et al., 2022*; *Yilmaz et al., 2018*). Therefore, the development of novel pharmaceuticals to treat male infertility caused by oligospermia holds substantial practical significance.

Ursolic acid (ULA, 3b-hydroxy-urs-12-en-28-oic acid, Fig. 1) is a naturally occurring pentacyclic triterpenoid carboxylic acid compound that was first extracted and identified from the apple peel in 1920s (*Khwaza, Oyedeji & Aderibigbe, 2020*). Current research has found that ULA is also widely present in various medicinal plants, vegetables, and fruits (*Samivel et al., 2020*; *Woźniak, Skąpska & Marszaek, 2015*). Studies have shown that ULA has multiple pharmacological properties such as anticarcinogenic (*Feng & Su, 2019*; *Panda, Thangaraju & Lokeshwar, 2022*; *Yin et al., 2018*), antiviral (*Al-Kuraishy, Al-Gareeb & Batiha, 2022*; *Tohmé et al., 2019*), anti-inflammatory (*Kashyap, Tuli & Sharma, 2016*; *Luan et al., 2022*), antihyperglycemic (*Tam et al., 2016*; *Wen et al., 2008*), and antioxidant (*Habtemariam, 2019*; *Srinivasan et al., 2020*). However, the effects and mechanism of ULA on oligospermia have not been reported.

Busulfan (BUS), a chemotherapeutic drug used to treat chronic myelogenous leukemia, can cause male reproductive system damage through the disruption of spermatogenesis and lead to oligospermia (*Meistrich, 2009*; *Zhao et al., 2020*). Motor proteins, which are considered key participants in a variety of processes during spermatogenesis and male infertility has been observed in animal models lacking several different motor proteins (*Ma, Wang & Yang, 2017*). In this study, we investigated the effects of ULA on BUS-induced oligospermia in mouse models. At the same time, we also explored the underlying mechanism of action of ULA on oligospermia. Results showed that ULA ameliorated oligospermia in BUS-induced mice by attenuating oxidative stress and promoting cytoskeleton remodeling *via* the motor proteins pathway. Our research will provide new insights for the treatment of male infertility caused by oligospermia with ULA.

## MATERIALS AND METHODS

### Materials

ULA (purity > 98%) was purchased from Aladdin (Shanghai, China). BUS was purchased from Sigma-Aldrich (B1170000, St. Louis, MO, USA). Hematoxylin and eosin (HE) was obtained from Biosharp (BL700B, China). $\alpha$-tubulin antibody was purchased from Cell

**Figure 1** Molecular structure of ursolic acid.

Signaling Technology (2144S, Japan). Actin-Tracker Green-488 and Alexa Fluor 555-labeled donkey anti-rabbit IgG were purchased from Beyotime Biotechnology (C2201S, A0453, Jiangsu, China). Antifade mounting medium for fluorescence (with DAPI) was purchased from Biosharp (BL739A, Hefei City, China). Trizol reagent was purchased from Vazyme (R401-01, Nanjing, China).

## Animals

Male ICR mice ($n = 50$, 8 weeks old, $30 \pm 2$ g) were purchased from the Laboratory Animal Center of Nantong University (Nantong, China). All mice were housed at a standard cage size of 5 in standard laboratory environmental conditions (room temperature 22–24 °C, 12/12-h light/dark cycle) with free access to food and water. All mice received a single intraperitoneal injection and gavage for four weeks. There is no surgical procedure and no analgesia or anesthesia is required. After administration, all mice were euthanized using cervical dislocation, and no mice were euthanized prior to the planned end of the experiment. There are no surviving animals at the conclusion of the experiment All procedures have been approved by the Laboratory Animal Center of Nantong University, approval ID: IACUC20230415-1001.

## Experimental groups, treatment, and sample preparation

A total of fifty mice were randomly assigned into five distinct groups, with the allocation known only to the principal investigator: a normal group, a BUS group, and three BUS

plus ULA groups at dosages of 10 mg/kg, 30 mg/kg, and 50 mg/kg, respectively. The BUS solution was prepared using a 5% dimethyl sulfoxide (DMSO) in water, while ULA was suspended in a 5% DMSO in corn oil vehicle. To establish the oligospermia model, BUS was administered *via* intraperitoneal injection at a dosage of 30 mg/kg body weight. The normal group received an equivalent dose of the 5% DMSO water solution (*Wang et al., 2010*). After a two-week period post-BUS administration, the mice in the ULA groups were administered different concentrations of ULA daily *via* oral gavage for a duration of four weeks. The normal and BUS groups were given an equivalent dose of the 5% DMSO in corn oil solution. At the end of the treatment period, each mouse was weighed. Following this, the animals were euthanized humanely under anesthesia. After euthanasia, blood samples were collected, and organs such as the testes and epididymis were excised for further experimental analysis.

## Analysis of epididymal sperm concentration and motility

Immediately after euthanasia, the entire epididymis was removed and the cauda epididymis was minced in a small dish containing two mL of Tyrode's solution and incubated at 37 °C for 15 min with gentle shaking to expel sperm. Sperm concentration and motility parameters were measured using a computer-assisted sperm analysis system (Hamilton Thorne CEROS II).

## Histology and immunofluorescent

Fixation in 4% paraformaldehyde was performed for the testes and epididymis for 24 h, followed by soaking in 10%, 20%, and 30% sucrose for an overnight period. Thin cryosections were made using a frozen-stage microtome and were stained with Hematoxylin-eosin (HE) using standard procedures, and observed under light microscopy. For immunofluorescence, after the sections were permeabilized, the primary antibody $\alpha$-tubulin was applied overnight, followed by the secondary antibody at room temperature for one hour. F-actin was stained according to the manufacturer's instructions, and observed using a fluorescence microscope (Leica TCS-SP8 LSM, Leica, Wetzlar, Germany).

## Serum hormone levels

Blood was collected by cardiac puncture with a heparinized syringe, followed by immediate collection of serum by centrifugation at 3,000 g, 4 °C for 10 min. Samples were stored at −80 °C for further analysis. Serum levels of testosterone (T), follicle-stimulating hormone (FSH) and luteinizing hormone (LH) were measured by ELISA kits according to the manufacturer's instructions. The sensitivity of the mouse T, FSH, and LH assays was 0.10 ng/mL, 0.55 ng/mL, and 0.07 ng/mL, respectively.

## ROS and MDA levels in serum

The levels of reactive oxygen species (ROS) and malondialdehyde (MDA) in serum were detected by ELISA kits, and the operation was performed strictly according to the instructions of the kit. The sensitivities of the mouse ROS and MDA assays were 5 IU/mL and 0.075 nmol/L, respectively.

## RNA sequencing and data analysis

The extraction of total RNA from the testicular tissue was conducted utilizing the Trizol reagent, following the protocol provided by the supplier. The RNA was then characterized and its concentration was measured using a NanoDrop spectrophotometer along with a 2100 Bioanalyzer. Subsequently, the PCR amplicons were employed to construct complementary DNA (cDNA) libraries. These libraries were subjected to sequencing using the BGISEQ-500/MGISEQ-2000 platforms, which are products of BGI-Shenzhen, China. The initial data underwent comprehensive quality assessment and refinement through the Fastp tool. The refined data sets were mapped against the Mus musculus reference genome GRCm38 using the STAR software, enabling the calculation of mapping efficiency for each sample. The FeatureCounts software was employed to ascertain the gene counts across the samples. With two biological replicates and one technical replicate, the experimental design's statistical power was computed to be 0.79 using RNASeqPower, considering a sequencing depth of 8, a sample size of 4, a coefficient of variation of 0.02, an effect size of 2, and an alpha level of 0.05 (https://rodrigo-arcoverde.shinyapps.io/rnaseq_power_calc/). Subsequently, the ClusterProfiler toolkit was utilized to analyze the differential gene expression patterns among the samples. The genes that showed significant variation were further analyzed using Gene Ontology (GO) terms and Kyoto Encyclopedia of Genes and Genomes (KEGG) pathways. This deeper analysis aimed to reveal the biological implications of these genes and their roles in various metabolic and cellular processes.

## Quantitative real-time PCR

Expression of relevant genes was further assessed by quantitative real-time PCR. RNA was used to synthesize cDNA using a reverse transcription kit. cDNA was estimated using a NanoDrop 2000/2000C spectrophotometer. The $\beta$-actin gene was used as an internal control. PCR reactions were carried out in triplicate on the CFX Connect system (Bio-RAD, Hercules, CA, USA) using ChamQ Universal SYBR qPCR Master Mix (Q711, Nanjing, Vazyme). The fold change of the expression of the mRNA ($2^{-\Delta\Delta Ct}$) was used in the data analysis.

## Data availability statement

The data supporting the findings of this study are available within the article and its supplemental materials. RNA sequencing data files has been deposited in the NCBI Gene Expression Omnibus (GEO) under accession number GSE251662, which is publicly accessible at https://www.ncbi.nlm.nih.gov/geo/.

## Statistical analysis

GraphPad Prism 8.1 was chosen for analysis, and the data were all assigned as mean $\pm$ SEM. One-way analysis of variance was applied to investigate differences between multiple experimental groups, and student's $t$-tests were employed to examine differences between two experimental groups. It was determined to be statistically significant when $P < 0.05$.

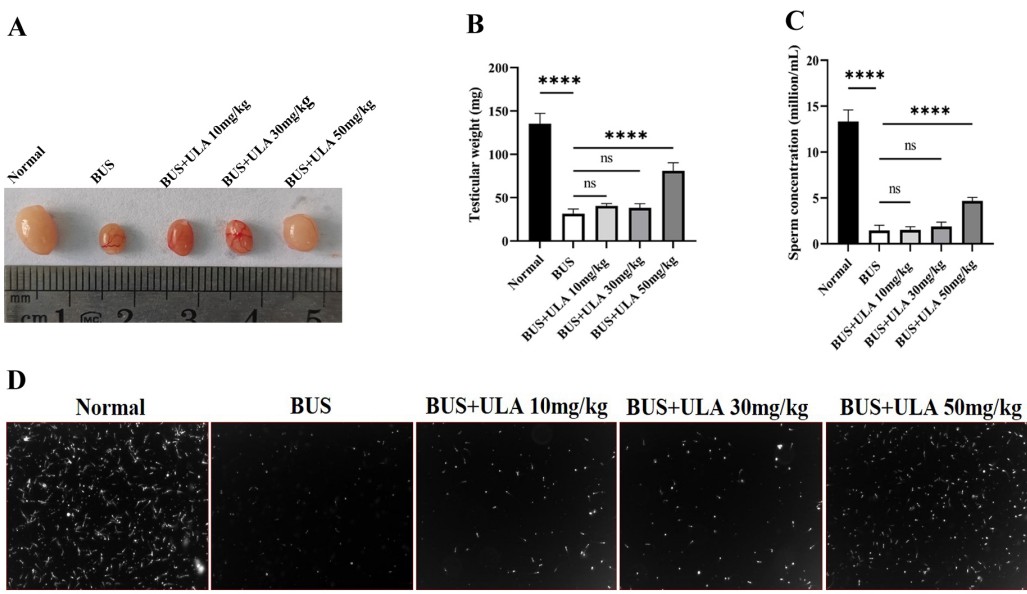

**Figure 2** **Reproductive organ index and sperm parameters.** (A) Morphology of the testes from 8-week-old mice after treatment with solution, BUS 30 mg/kg, BUS+ULA 10 mg/kg, BUS+ULA 30 mg/kg, BUS+ULA 50 mg/kg. (B) Testicular weight. (C) Sperm concentration. (D) Morphology of sperms from epididymis. ****$P < 0.0001$. Each column represents the mean ± SEM, $n = 6$.

## RESULTS

### ULA alleviates reproductive damage induced by BUS in oligospermia mice

As shown in Fig. 2, compared with the normal group, the testicular weight and sperm concentration of mice treated with BUS decreased significantly, showing a typical oligospermia model. However, after giving ULA at varying concentrations, we found that the testicular weight and sperm concentration of oligospermia mice increased statistically significantly in the 50 mg/kg dosage group. These findings demonstrate the successful mitigation of reproductive system impairment induced by BUS through the administration of ULA.

### ULA promotes morphological recovery of testis and epididymis in oligospermia mice

The administration of BUS resulted in the loss of immature male cells, a reduction in both luminal diameter and vacuolization in immature male cells and seminiferous tubules, as well as significant destruction of peritubular mesenchymal cells. After treatment with ULA, the presence of regenerated germ cells was observed (Fig. 3A). The testes have the responsibility of producing sperm, which undergo maturation and acquire the capacity for fertilization within the epididymis (Kistler et al., 1975). Hematoxylin-eosin (HE) staining was employed to observe epididymis structure and integrity. In the BUS group, epithelial cells disintegrated and arranged disorganized, formed large number of vacuoles, epididymis wall became thicker, and the number of spermatozoa in the lumen of epididymis tail

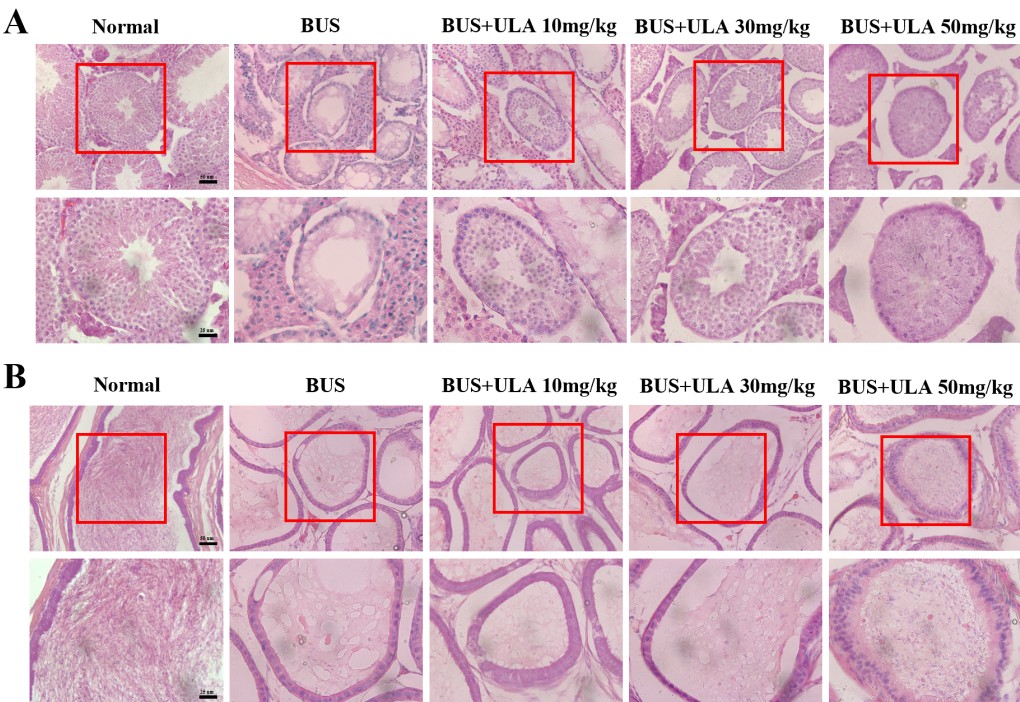

**Figure 3** **Histomorphometric analysis of mice testis and epididymis.** (A) HE staining shows the morphology of testis (20X). (B) HE staining displays the morphology of the epididymis tail (20X). Scale bar: 50 μm.

decreased. However, after ULA treatment, epididymal duct and its epithelial cells were partially restored, and the number of spermatozoa in theca epididymis increased (Fig. 3B).

## ULA restores cytoskeletal disorder in BUS-induced oligospermia mice

Microtubules and microfilaments are two types of cytosolic fibers that make up the cytoskeleton (*Westermann & Weber, 2003*). By employing immunofluorescence (IF) staining methods for $\alpha$-tubulin and F-actin, we observed changes in the cytoskeleton distribution pattern after different treatments. Frozen section samples were subjected to staining with $\alpha$-tubulin antibody and phalloidin at the specified times, and subsequently photographed under a fluorescent microscope to compare the fluorescence intensity. The images of the stained cells are shown in Fig. 4. In the normal group, cytoskeleton was detected at the lamina propria, exhibiting concentrated and track-like signals. Conversely, in the BUS group, the cytoskeleton tracks appeared truncated, more discrete and flocculated. Interestingly, the damage caused by BUS was significantly recovered after the use of ULA, as evidenced by the restoration of clear 'track-like signals' within the cytoskeleton.

## ULA promotes serum sex hormone secretion in oligospermia mice

The levels of sex hormones in serum serve as vital assessment parameters of reproductive system status. Testosterone (T) level is an important indicator of male reproductive health. Compared with the normal group, the administration of BUS resulted in a down-regulation

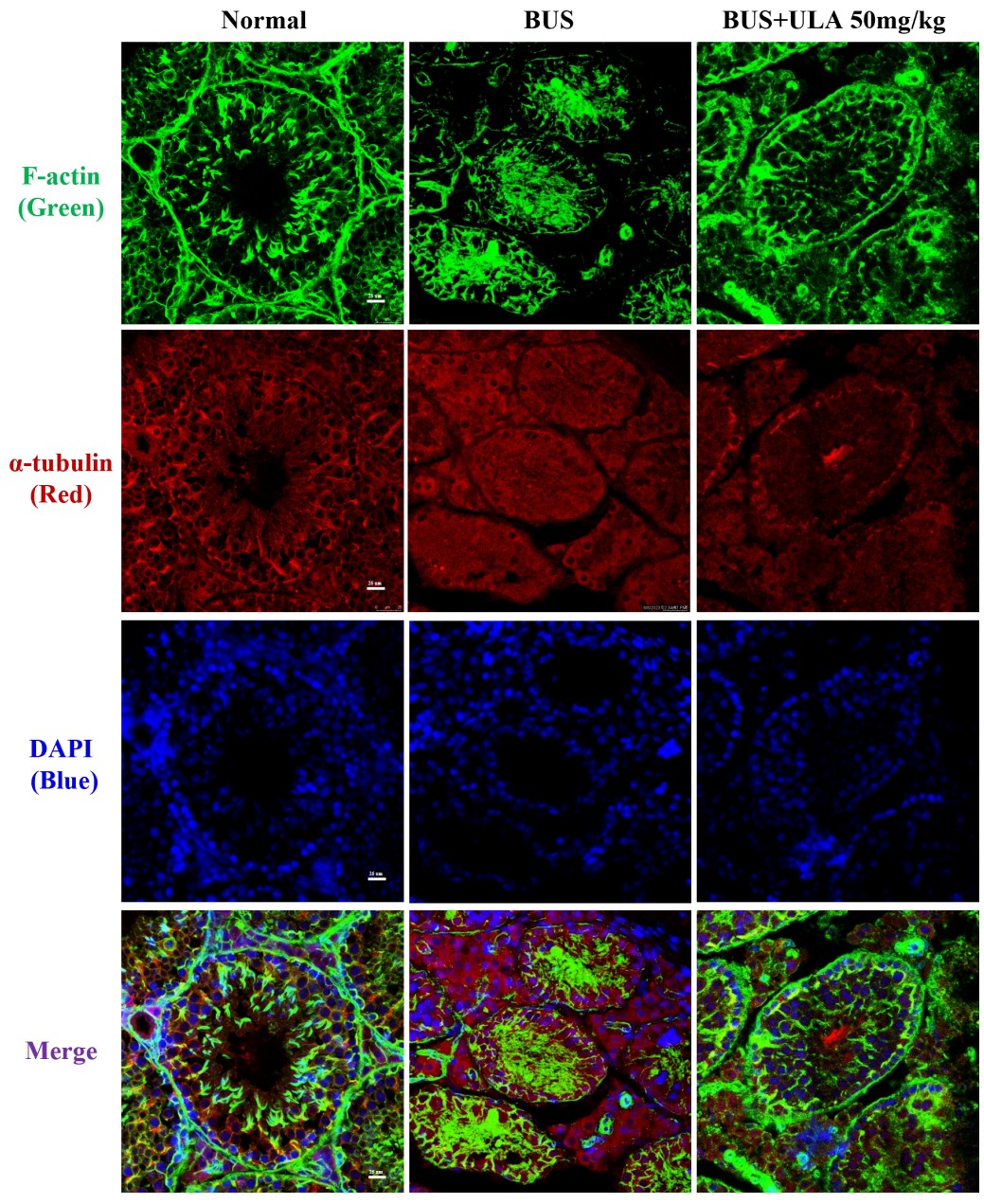

**Figure 4** Cytoskeleton changes detected by IF in each group.

of T, luteinizing hormone (LH) and follicle-stimulating hormone (FSH) levels (Fig. 5). Conversely, ULA treatment exhibited an upregulation in serum T, FSH and LH levels. These observations indicate that the damage to gonadal tissue induced by BUS can be ameliorated by ULA through regulating the secretion of gonadal hormones.

## ULA inhibits the oxidative stress in oligospermia mice

Lipid peroxidation in tissues can be assessed by measuring the tissue MDA content (*Grimes Jr et al., 1975*). As illustrated in Fig. 6, BUS treatment led to a substantial elevation in MDA

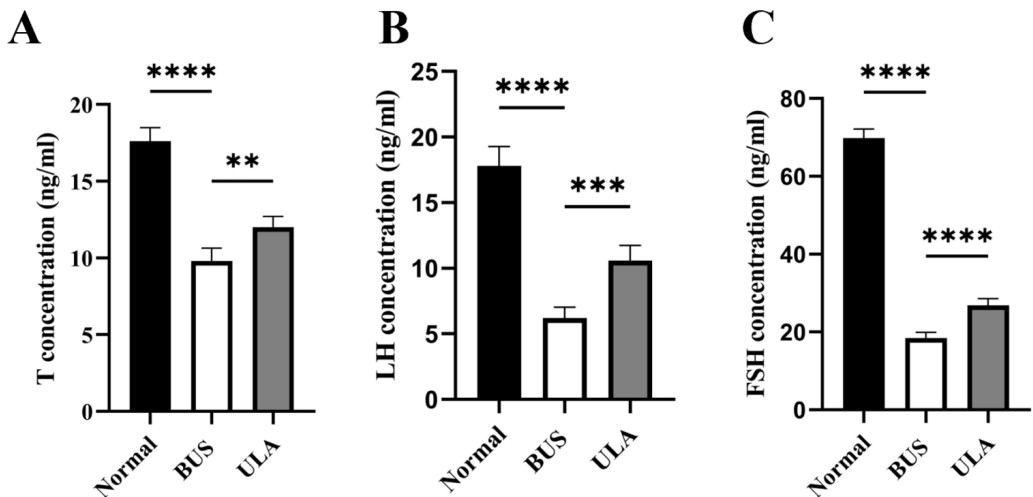

**Figure 5** **Effect of ULA on sex hormone levels.** (A) T concentration. (B) LH concentration. (C) FSH concentration. $**P < 0.01$, $***P < 0.001$, $****P < 0.0001$. Each column represents the mean $\pm$ SEM, $n = 6$.

and ROS levels in the testes compared to the normal group. The heightened oxidation levels indicated that oxidative stress which tends to be considered as one of the primary factors contributing to oligospermia occurs in oligospermia mice, resulting in reproductive damage. However, a reduction in both MDA and ROS levels was observed after ULA treatment, suggesting that ULA treatment mitigated oxidative damage caused by BUS.

## Differential analysis of gene expression in the testis between BUS and BUS+ULA groups

To investigate the underlying mechanism of ULA in the treatment of oligospermia, RNA sequencing analysis was conducted to identify differentially expressed genes in the mouse testes between the BUS and BUS + ULA groups. After 4 weeks of ULA treatment, a total of 2,157 genes were significantly up-regulated, while 3,871 genes were significantly down-regulated (Figs. 7A–7B). To better understand the biological functions associated with these differentially expressed genes, GO classification analysis based on RNA sequencing data was performed. The top eight enriched GO biological processes are visualized in Fig. 7D. In the GO terminology, differentially expressed genes involved in "cilium movement", "sperm motility", "cell motility", and "spermatid development" were shown to be closely related to the effect of ULA on spermatogenesis. Additionally, KEGG enrichment analysis was conducted to explore the specific mechanisms of ULA treatment. KEGG enrichment analysis revealed that the expression of genes increased most significantly in ULA group was found to be enriched in the motor proteins (Fig. 7E, yellow background) which perform essential functions in cellular processes such as transport and secretion. We verified the sequencing results of motor proteins by qPCR (Fig. 7C). The movement of molecules within cells is facilitated by motor proteins which rely on energy derived from the hydrolysis of adenosine triphosphate (ATP) for their power source. Among the differentially expressed mRNAs associated with germ cell development, RNA sequencing

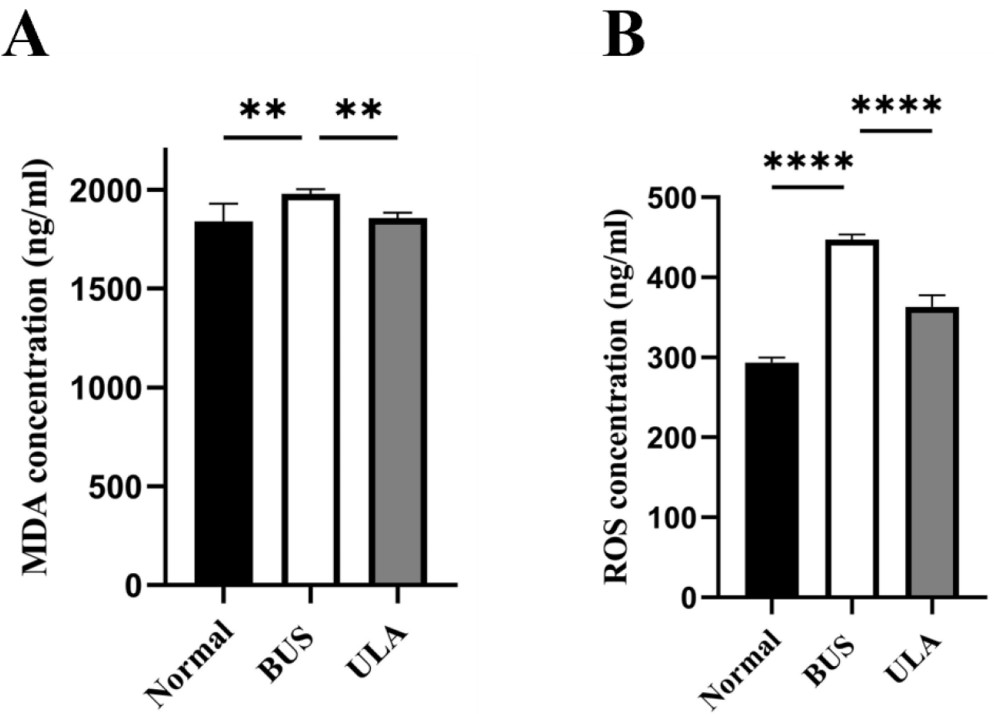

**Figure 6 Effects of ULA on oxidative indices in tissues.** (A) MDA concentration. (B) ROS concentration. **$P < 0.01$, ****$P < 0.0001$. Each column represents the mean $\pm$ SEM, $n = 6$.

demonstrated a significant upregulation (>1.5-fold change, $p < 0.05$) in the expression of relevant genes following ULA treatment compared to the BUS group.

## DISCUSSION

At present, oligospermia patients exhibit diverse clinical symptoms, and the mechanism underlying non-obstructive oligospermia remains unclear. However, low level of T and high level of ROS are commonly observed. It has been reported that BUS administration could significantly increase ROS in spermatogonia and inhibit its differentiation and proliferation (*Jing et al., 2023*). ROS is a double-edged sword in spermatogenesis, high level of ROS could lead to oxidative stress and cell death. Therefore, antioxidant therapy has emerged as a potential approach for the treatment of non-obstructive oligospermia (*Aitken & Drevet, 2020*). Consequently, depleting endogenous germ cells by single intraperitoneal injection of BUS has been proven to be an effective strategy for creating animal models of male infertility due to oligospermia.

ULA is a pentacyclic triterpenoid which is similar to steroid hormones in plants and mammals that are primarily synthesized through the MVA pathway (*Ikeda, Murakami & Ohigashi, 2008*). It is classified as a class IV complex, showing poor oral bioavailability, low solubility and limited intestinal permeability, according to the Biopharmaceutics Classification System. However, ULA exhibits surprisingly strong pharmacodynamic

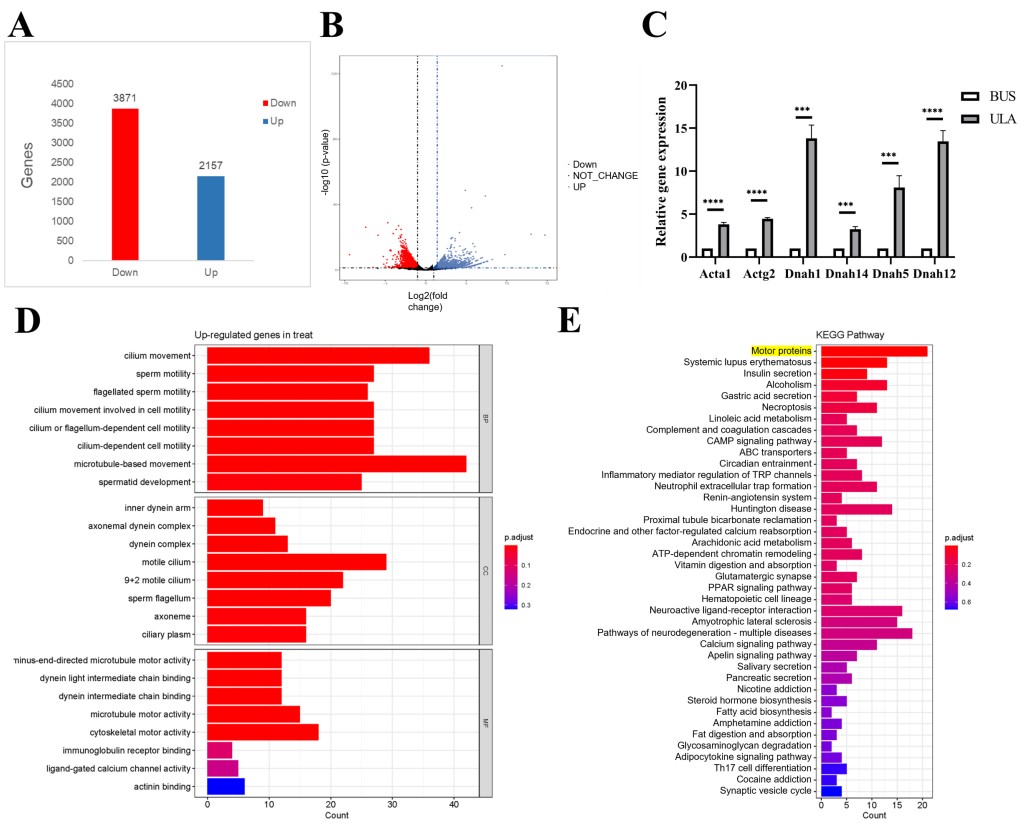

**Figure 7 High-throughput sequencing.** (A) The number of differentially expressed genes. (B) Volcano map showing differentially expressed genes between oligospermia and ULA treatments. (C) RT-qPCR of motor proteins. (D) The GO enrichment analysis. (E) The KEGG enrichment analysis. ***$P < 0.001$, ****$P < 0.0001$.

properties and bioactivities (*Sun et al., 2020*). The absorption of ULA through the intestinal tract is primarily accomplished *via* passive diffusion, although some evidence suggests that it may also be actively transported. ULA likely serves as a substrate for the transmembrane protein called Permeability Glycoprotein 1, which plays a crucial role in this process. Once inside the body, ULA is quickly broken down and eliminated by the liver, contributing to its limited bioavailability (*Hua, Fang & Hua, 2012*). The toxic dose is higher than 1,000 mg/kg/day in rats and at single oral doses up to 1,000 mg in healthy adult human volunteers of ULA also found no serious adverse event (*Geerlofs et al., 2020*; *Hirsh et al., 2014*). These findings suggest that ULA has low toxicity in both rodents and humans. Low water solubility and low bioavailability are common problems of ULA and most of its derivatives, which seriously limit their utilization. There are several different ways to improve ULA bioavailability, including adding salt ions to increase its water solubility and improving its absorption in the stomach and inhibiting metabolism in the liver. It has been reported that a high-fat diet can promote the absorption of triterpenes (*Furtado et al., 2017*). In brief, increasing its bioavailability should be the tendency of the alteration of ULA.

ULA has been reported that has a protective effect on LPS-Induced asthenozoospermia *via* Bcl-2/Bax apoptosis signaling pathway (*Sun et al., 2021*). In this study, ULA partially reversed BUS-induced oligospermia. The testicular spermatogenic cells at all levels and interstitial cells destroyed by BUS were restored accompanied with the reconstruction of cytoskeleton after ULA treatment. The level of serum sex hormone decreased due to the destruction of testicular tissue, and the level of ROS and MDA in oligospermia mice under oxidative stress was adjusted. However, the improvement of ULA is limited while the sperm motility didn't get recovery. We hypothesized that ULA's abundant pharmacological activity played a role, but its poor bioavailability limited its efficacy. There was no statistically significant change in sperm motility in oligospermia mice after administration.

The RNA sequencing and GO enrichment analysis indicated that the biological processes of "cilium movement", "sperm motility", "cell motility " and "spermatid development" were closely related with the effect of ULA on spermatogenesis which is consistent with "motor proteins" shown in KEGG enrichment analysis. Motor proteins consists of three superfamilies myosin motors, kinesin motors and dynein motors. They power directed movements on microtubules or actin filaments (*Mouriño Pérez et al., 2016*; *Subramanian & Kapoor, 2012*). Kinesin motors and dynein motors are considered key participants in a variety of processes during spermatogenesis. Observations from various animal models lacking specific kinesin motors have highlighted male infertility; thus, kinesins should be considered an integral part of discussions on male reproductive issues (*Ma, Wang & Yang, 2017*). Kinesin and dynein motors move on microtubules, whereas myosin motors move on actin filaments. Motor proteins use the chemical energy generated by the hydrolysis of ATP to drive themselves along microtubules or microfilaments and perform fundamental biological processes, such as intracellular transport, protein degradation and energy production (*Liu, Chistol & Bustamante, 2014*; *Vale, 2003*). Our study showed that the damage in the testis of oligospermia mice was characterized by obvious cytoskeletal disorder and microfilament disorganization while microtubules and microfilaments gained reconstruction after the use of ULA accompanied with the upregulating of motor proteins.

## CONCLUSION

In summary, we discovered that ULA ameliorates BUS-induced oligospermia in mice by upregulating motor proteins to protect cytoskeleton from alkylating agents. ULA improves the serum sexual hormone secretion and down-regulates the degree of oxidative damage in oligospermia mice. After improving its bioavailability, ULA has the potential to serve as a potent therapeutic agent for non-obstructive oligospermia treatment.

### Funding

This study was supported by the National Natural Science Foundation of China (81901528 to Xiaorong Wang), the Jiangsu Health Innovation Team Program (2020), the Nantong

University Clinical Medicine Specialized Research Fund Project (2023JY008 to Xiaorong Wang) and the Postgraduate Research & Practice Innovation Program of Jiangsu Province (KYCX24_3562 to Taowen Ye). The funders had no role in study design, data collection and analysis, decision to publish, or preparation of the manuscript.

### Grant Disclosures

The following grant information was disclosed by the authors:

National Natural Science Foundation of China: 81901528.

Jiangsu Health Innovation Team Program (2020).

Nantong University Clinical Medicine Specialized Research Fund Project: 2023JY008.

Postgraduate Research & Practice Innovation Program of Jiangsu Province: KYCX24_3562.

### Competing Interests

The authors declare there are no competing interests.

### Author Contributions

- Jin Dong conceived and designed the experiments, performed the experiments, prepared figures and/or tables, and approved the final draft.
- Taowen Ye performed the experiments, prepared figures and/or tables, and approved the final draft.
- Yanli Dong analyzed the data, authored or reviewed drafts of the article, and approved the final draft.
- Jie Hui analyzed the data, prepared figures and/or tables, and approved the final draft.
- Xiaorong Wang conceived and designed the experiments, authored or reviewed drafts of the article, and approved the final draft.

### Animal Ethics

The following information was supplied relating to ethical approvals (i.e., approving body and any reference numbers):

Laboratory Animal Center of Nantong University provided full approval for this research (IACUC20230415-1001).

### DNA Deposition

The following information was supplied regarding the deposition of DNA sequences:

RNA sequencing data are available at NCBI Gene Expression Omnibus (GEO): GSE251662.

### Data Availability

Raw data are available in the Supplemental Files.

### Supplemental Information

Supplemental information for this article can be found online at http://dx.doi.org/10.7717/peerj.17691#supplemental-information.

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
