# Peer review of "Ursolic acid attenuates oligospermia in busulfan-induced mice by promoting motor proteins"

_PeerJ, doi:10.7717/peerj.17691_

## Round 0.1 · original submission · Minor Revisions

Please address concerns of all reviewers and mend manuscript accordingly.

Reviewer 1 ·

Basic reporting

no comments

Experimental design

no comments

Validity of the findings

no comments

Additional comments

1. Please provide more insight about the role of motor proteins kinesin, dynein and microtubules in oligospermia in the discussion.
2. Line 82 please provide reference for the motor protein pathway affecting oligospermia.
3. There is increase in sperm count and not in its motility, does ursolic acid improves the fertility in oligospermia.
4. Line 113 please write body weight instead of b.w.
5. Please mention that the control group received only DMSO.

Reviewer 2 ·

Basic reporting

good

Experimental design

good

Validity of the findings

good

Additional comments

The manuscript entitled "Ursolic acid attenuates oligospermia in busulfan-induced mice by promoting motor proteins" delineates compelling biological data concerning the favorable influence of ursolic acid on the sperm parameters. I think the authors have, in general, done a good job of building this case. However, there are some minor issues necessitating further elucidation. I propose manuscript acceptance following minor revisions.

1. How were the doses of busulfan chosen to induce the oligospermia model? Any report about it?
2. The section "Experimental Groups, Treatment, and Sample Preparation" within Materials and Methods requires more comprehensive detailing.
3. In Figure 7C, qPCR results should be denoted with statistical significance markers on the graph.
4. Reference citation at Line 369-371:
"Hirsh S, Huber L, Zhang P, Stein R, and Joyal S. 2014. A single ascending dose, initial clinical pharmacokinetic and safety study of ursolic acid in healthy adult volunteers (1044.6). The FASEB Journal 28:1044.6. https://doi.org/10.1096/fasebj.28.1_supplement.1044.6";
should be
"Hirsh S, Huber L, Zhang P, Stein R, and Joyal S. 2014. A single ascending dose, initial clinical pharmacokinetic and safety study of ursolic acid in healthy adult volunteers (1044.6). The FASEB Journal 28:1044.6".
5. Figure 3 and 4 lack corresponding scale bars. It is recommended to add them.
6. Line 74/79/89 "…ursolic acid…"should be abbreviated as "…ULA…".

Reviewer 3 ·

Basic reporting

The manuscript titled: " Ursolic Acid attenuates oligospermia in busulfan-induced mice by promoting motor proteins" is a well-composed article with multiple line of evidence on how the natural compound Ursoloic Acid (ULA) may have potential in reducing chemotherapy -induced oligospermia in men. The premise of the hypothesis, experimental design and presented results are well organized and the results are clearly described in text as well as in Figures. The work article has sufficient amount of background information on how Buslfan, a chemotherapy drug used to treat multiple myeloma, causes oligospermia in mice and that can be traced to high levels of testicular tissue degradation, reduction in testosterone and FSH secretion and as a result production of defective sperms deficient in motility. The authors have provided a thorough set of experiments including RNA sequencing to analyze macromolecular imbalanced caused by BUS damage and their normalization with ULA. The references are current and sufficient information is presented.
Within the context of the subject addressed in this article, few significant weaknesses maybe identified. However, in a larger context of scientific rigor and mechanistic understanding there are few weaknesses which are not fatal, but can be addressed in the future. These are: 1. Lack of the levels of ULA in serum or in tissues (those analyzed in this MS), No clear explanation why MDA levels are decreased in ULA treated samples but not ROS concentration, and very limited analysis of RNA-sequence data. However, providing Q-PCR verification of major altered genes is an advantage.

Experimental design

The experimental design is sound and extensive. No major weaknesses identified. Treating the mice with multiple levels of ULA is a good choice, however, the results show the data were statistically significant only in the highest dose. The authors could have provided some serum toxicity data since they collected the sera following four weeks of treatment. A minor comment, authors could have used UA instead of ULA as the more accepted abbreviation for UA.
In the experimental design, inclusion of ferticilty of BUS treated mice with and without ULA post treatment (4 weeks after) would have provided a significant boost to the scientific rigor of the manuscript and much needed evidence for therapeutic potential of ULA in treating oligospermia.

Validity of the findings

All findings are valid as they are rigorously validated with statistical testing and using multiple line of testing. Overall this is an interested manuscript and should have a significant impact on the potential in the future. Two problems remains. One is the ULA treatment did not elevate motility as effectively it did with restoring T levels. The second is the authors did not compare any other drug that has been used to ameliorate oligospermia in clinic. This would have probably given a wide perspective for ULA's prospect.

Additional comments

1. If possible the authors should measure the level of ULA in serum using the well-established HPLC -MS method published and referenced in Ref. by Sun Q et al. and other referenced articles.
2. The authors did not compare the efficacy of ULA with an established drug on treating Oligospermia (such as long-acting testoserone pellets).

---

## Round 0.2 · accepted · Accept

All issues pointed by the reviewers were adequately addressed and the manuscript was amended accordingly. Therefore, revised manuscript is acceptable now.